# GENERALIZED ZERO-SHOT ICD CODING

## ABSTRACT

The International Classification of Diseases (ICD) is a list of classification codes for the diagnoses. Automatic ICD coding is a multi-label text classification task with noisy clinical document inputs and extremely long-tailed label distribution, making it difficult to perform fine-grained classification on both frequent and zero-shot codes at the same time. In this paper, we propose a latent feature generation framework for generalized zero-shot ICD coding, where we aim to improve the prediction on codes that have no labeled data without compromising the performance on seen codes. Our framework generates semantically meaningful features for zero-shot codes by exploiting ICD code hierarchical structure and a novel cycle architecture that reconstructs the relevant keywords. To the best of our knowledge, this is the first adversarial generative model for the generalized zero-shot learning on multi-label text classification. Extensive experiments demonstrate the effectiveness of our approach. On the public MIMIC-III dataset, our methods improve the F1 score from nearly 0 to $20.91\%$ for the zero-shot codes, and increase the AUC score by 3% (absolute improvement) from previous state of the art.

## 1 INTRODUCTION

In healthcare facilities, clinical records are classified into a set of International Classification of Diseases (ICD) codes that categorize diagnoses. ICD codes are used for a wide range of purposes including billing, reimbursement, and retrieving of diagnostic information. Automatic ICD coding (Stanfill et al., 2010) is in great demand as manual coding can be labor-intensive and error-prone. ICD coding is a multi-label text classification task with a long-tailed class label distribution. Majority of ICD codes only have a few or no labeled data due to the rareness of the disease. In the medical dataset MIMIC III (Johnson et al., 2016), among 17,000 unique ICD-9 codes, more than $50\%$ of them never occur in the training data. It is extremely challenging to perform fine-grained multi-label classification on both codes with labeled data (seen codes) and zero-shot (unseen) codes at the same time. Automatic ICD coding for both seen and unseen codes fits into the generalized zero-shot learning (GZSL) paradigm (Chao et al., 2016), where test examples are from both seen and unseen classes and we classify them into the joint labeling space of both types of classes. Nevertheless, current GZSL works focus on visual tasks (Xian et al., 2017; Felix et al., 2018; Liu et al., 2019). The study of GZSL for multi-label text classification is largely under-explored.

Modern automatic ICD coding models (Mullenbach et al., 2018; Rios & Kavuluru, 2018) can accurately assign frequent ICD codes while perform poorly on zero-shot codes. To resolve this discrepancy, we aim to improve the predictive power of existing models on zero-shot codes by finetuning the models with synthetic latent features. To generate semantically meaningful features, we exploit the domain knowledge about ICD codes. The official ICD guidelines provide each code a short text description and a hierarchical tree structure on all the ICD codes (ICD-9 Guidelines). We propose AGMC-HTS, an **A**dversarial **G**enerative **M**odel **C**onditioned on code descriptions with **H**ierarchical **T**ree **S**tructure to generate synthetic feature. As illustrated in Figure 1, AGMC-HTS consists of a generator to synthesize code-specific latent features based on the ICD code descriptions, and a discriminator to decide how realistic the generated features are. To guarantee the semantic consistency between the generated and real features, AGMC-HTS reconstructs the keywords in the input documents that are relevant to the conditioned codes. To further facilitate the feature synthesis of zero-shot codes, we take advantage of the hierarchical structure of the ICD codes and encourage the zero-shot codes to generate similar features with their nearest sibling code. The ICD coding models are finetuned on the generated features to achieve more accurate prediction for zero-shot codes.

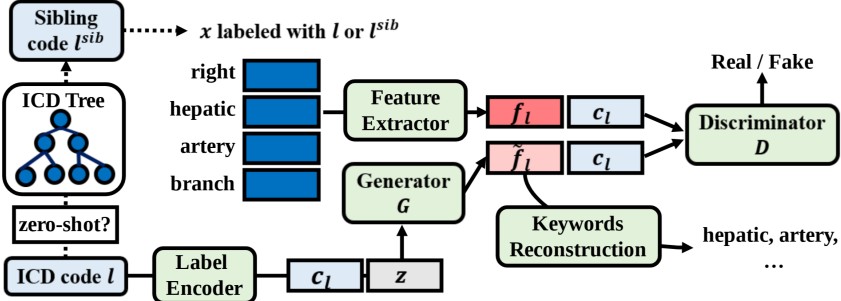

Figure 1: Overview of AGMC-HTS. The generator synthesizes features for an ICD code and the discriminator decides how realistic the input feature is. For a zero-shot ICD code, the discriminator distinguishes between the generated features and the real features from the data of its nearest sibling in the ICD hierarchy. The generated features are further used to reconstruct the keywords in the input documents to preserve semantics.

The contributions of this paper are summarized as follows: 1) To the best of our knowledge, we propose the first adversarial generative model for the generalized zero-shot learning on multi-label text classification. AGMC-HTS generates latent features conditioned on the code descriptions and finetunes the zero-shot ICD code assignment classifiers. 2) AGMC-HTS exploits the hierarchical structure of ICD codes to generate semantically meaningful features for zero-shot codes without any labeled data. 3) AGMC-HTS has a novel pseudo cycle generation architecture to guarantee the semantic consistency between the synthetic and real features by reconstructing the relevant keywords in input documents. 4) Extensive experiments demonstrate the effectiveness of our approach. On MIMIC-III dataset, our methods improve the F1 score from nearly 0 to $20.91\%$ for the zero-shot codes and AUC score by $3\%$ (absolute improvement) from previous state of the art. We also show that AGMC-HTS improves the performance on few-shot codes with a handful of labeled data.

## 2 RELATED WORK

**Automated ICD coding.** Several approaches have explored automatic assigning ICD codes on clinical text data (Stanfill et al., 2010). Mullenbach et al. (2018) proposed to extract per-code textual features with attention mechanism for ICD code assignments. Shi et al. (2017) explored character based short-term memory (LSTM) with attention and Xie & Xing (2018) applied tree LSTM with ICD hierarchy information for ICD coding. Most existing work either focused on predicting the most common ICD code or did not utilize the ICD hierarchy structure for prediction. Rios & Kavuluru (2018) proposed a neural network models incorporating ICD hierarchy information that improved the performance on the rare and zero-shot codes. The performance is evaluated in terms of the relative ranks to other infrequent codes. The model hardly ever assign rare codes in its final prediction as we show in Section 4.2, making it impractical to deploy in real applications.

**Feature generation for GZSL.** The idea of using generative models for GZSL is to generate latent features for unseen classes and train a classifier on the generated features and real features for both seen and unseen classes. Xian et al. (2018) proposed using conditional GANs to generate visual features given the semantic feature for zero-shot classes. Felix et al. (2018) added a cycle-consistent loss on generator to ensure the generated features captures the class semantics by using linear regression to map visual features back to class semantic features. Ni et al. (2019) further improves the semantics preserving using dual GANs formulation instead of a linear model. Previous works focus on vision domain where the features are extracted from well-trained deep models on large-scale image dataset. We introduce the first feature generation framework tailored for zero-shot ICD coding by exploiting existing medical knowledge from limited available data.

**Zero-shot text classification.** Pushp & Srivastava (2017) has explored zero-shot text classification by learning relationship between text and weakly labeled tags on large corpus. The idea is similar to Rios & Kavuluru (2018) in learning the relationship between input and code descriptions. Zhang et al. (2019a) introduced a two-phase framework for zero-shot text classification. An input is first determined as from a seen or an unseen classes before the final classification. This approach does not directly apply to ICD coding as the input is labeled with a set of codes which can include both seen and unseen codes. It is not possible to determine if the data is from a seen or an unseen class.

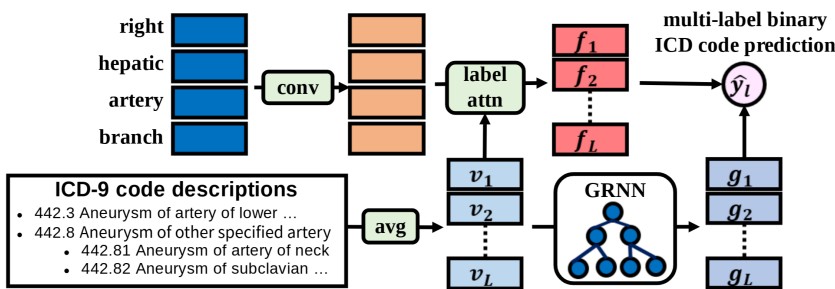

Figure 2: ZAGRNN as the feature extractor model . ZAGRNN extracts label-wise features and construct embedding of each ICD codes using GRNN. ZAGRNN makes a binary prediction for each code based on the dot product between graph label embedding and the label specific feature.

# 3 METHOD

## 3.1 OVERVIEW

The task of automatic ICD coding is to assign ICD codes to patient's clinical notes. We formulate the problem as a multi-label text classification problem. Let $\mathbb{L}$ be the set of all ICD codes and $L = |\mathbb{L}|$, given an input text, the goal is to predict $y_l \in \{0, 1\}$ for all $l \in \mathbb{L}$. Each ICD code $l$ has a short text description. For example, the description for ICD-9 code 403.11 is *"Hypertensive chronic kidney disease, benign, with chronic kidney disease stage V or end stage renal disease."* There is also a known hierarchical tree structure on all the ICD codes: for a node representing an ICD code, the children of this node represent the subtypes of this ICD code.

We focus on the generalized zero-shot ICD coding problem: accurately assigning code $l$ given that $l$ is never assigned to any training text (i.e. $y_l = 0$), without sacrificing the performance on codes with training data. We assume a pretrained model as a feature extractor that performs ICD coding by extracting label-wise feature $f_l$ and predicting $y_l$ by $\sigma(g_l^\top \cdot f_l)$, where $\sigma$ is the sigmoid function and $g_l$ is the binary classifier for code $l$. For the zero-shot codes, $g_l$ is never trained on $f_l$ with $y_l = 1$ and thus at inference time, the pretrained feature extractor hardly ever assigns zero-shot codes.

We propose to use generative adversarial networks (GAN) (Goodfellow et al., 2014) to generate $\tilde{f}_l$ with $y_l = 1$ by conditioning on code $l$. Figure 1 shows an overview of the generation framework. The generator $G$ tries to generate the fake feature $\tilde{f}$ given an ICD code description. The discriminator $D$ tries to distinguish between $\tilde{f}$ and real latent feature $f$ from the feature extractor model. After the GAN is trained, we use $G$ to synthesize $\tilde{f}_l$ and fine-tune the binary classifier $g_l$ with $\tilde{f}_l$ for a given zero-shot code $l$. Since the binary code classifiers are independently fine-tuned for zero-shot codes, the performance on the seen codes is not affected, achieving the goal of GZSL.

## 3.2 FEATURE EXTRACTOR

The pretrained feature extractor model is zero-shot attentive graph recurrent neural networks (ZA-GRNN) modified from zero-shot attentive graph convolution neural networks (ZAGCNN), which is the only previous work that is tailored towards solving zero-shot ICD coding (Rios & Kavuluru, 2018). We improve the original implementation by replacing the GCNN with GRNN and adopting the label-distribution-aware margin loss (Cao et al., 2019) for training. The detailed discussion of the performance gain by our modification is described Appendix E. Figure 2 shows the architecture of the ZAGRNN. At a high-level, given an input $x$, ZAGRNN extracts label-wise feature $f_l$ and performs binary prediction on $f_l$ for each ICD code $l$.

**Label-wise feature extraction.** Given an input clinical document $x$ containing $n$ words, we represent it with a matrix $X = [w_1, w_2, \ldots, w_n]$ where $w_i \in \mathbb{R}^d$ is the word embedding vector for the $i$-th word. Each ICD code $l$ has a textual description. To represent $l$, we construct an embedding vector $v_l$ by averaging the embeddings of words in the description.

The word embedding is shared between input and label descriptions for sharing learned knowledge. Adjacent word embeddings are combined using a one-dimension convolutional neural network (CNN) to get the n-gram text features $H = \text{conv}(X) \in \mathbb{R}^{N \times d_c}$. Then the label-wise attention

feature $a_l \in \mathbb{R}^d$ for label $l$ is computed by:

$$s_l = \text{softmax}(\tanh(H \cdot W_a^\top + b_a) \cdot v_l), \quad a_l = s_l^\top \cdot H \quad \text{for } l = 1, 2, \dots L$$

where $s_l$ is the attention scores for all rows in $H$ and $a_l$ is the attended output of $H$ for label $l$. Intuitively, $a_l$ extracts the most relevant information in $H$ about the code $l$ by using attention. Each input then has in total $L$ attention feature vectors for each ICD code.

**Multi-label classification.** For each code $l$, the binary prediction $\hat{y}_l$ is generated by:

$$f_l = \text{rectifier}(W_o \cdot a_l + b_o), \qquad \hat{y}_l = \sigma(g_l^\top \cdot f_l)$$

We use graph gated recurrent neural networks (GRNN) (Li et al., 2015) to encode the classifier $g_l$. Let $\mathcal{V}(l)$ denote the set of adjacent codes of $l$ from the ICD tree hierarchy and $t$ be the number of times we propagate the graph, the classifier $g_l = g_l^t$ is computed by:

$$g_l^0 = v_l, \quad h_l^t = \frac{1}{|\mathcal{V}(l)|} \Sigma_{j \in \mathcal{V}(l)} g_j^{t-1}, \quad g_l^t = \text{GRUCell}(h_l^t, g_l^{t-1}) \tag{1}$$

where GRUCell is a gated recurrent units (Chung et al., 2014) and the construction is detailed in Appendix A. The weights of the binary code classifier is tied with the graph encoded label embedding $g_l$ so that the learned knowledge can also benefit zero-shot codes since label embedding is computed from a shared word. The loss function for training is multi-label binary cross-entropy:

$$\mathcal{L}_{\text{BCE}}(y, \hat{y}) = -\sum_{l=1}^{L} [y_l \log(\hat{y}_l) + (1 - y_l) \log(1 - \hat{y}_l)] \tag{2}$$

As mentioned above, the distribution of ICD codes is extremely long-tailed. To counter the label imbalance issue, we adopt label-distribution-aware margin (LDAM) (Cao et al., 2019), where we subtract the logit value before sigmoid function by a label-dependent margin $\Delta_l$:

$$\hat{y}_l^m = \sigma(g_l^\top \cdot f_l - \mathbf{1}(y_l = 1)\Delta_l) \tag{3}$$

where function $\mathbf{1}(\cdot)$ outputs 1 if $y_1 = 1$ and $\Delta_l = \frac{C}{n_l^{1/4}}$ and $n_l$ is the number of training data labeled with $l$ and $C$ is a constant. The LDAM loss is thus: $\mathcal{L}_{\text{LDAM}} = \mathcal{L}_{\text{BCE}}(y, \hat{y}^m)$.

## 3.3 ZERO-SHOT LATENT FEATURE GENERATION WITH WGAN-GP

For a zero-shot code $l$, the code label $y_l$ for any training data example is $y_l = 0$ and the binary classifier $g_l$ for code assignment is never trained with data examples with $y_l = 1$ due to the dearth of such data. Previous works have successfully applied GANs for GZSL in the vision domain (Xian et al., 2018; Felix et al., 2018). We propose to use GANs to improve zero-shot ICD coding by generating pseudo data examples in the latent feature space for zero-shot codes and fine-tuning the code-assignment binary classifiers using the generated latent features.

More specifically, we use the Wasserstein GAN (Arjovsky et al., 2017) with gradient penalty (WGAN-GP) (Gulrajani et al., 2017) to generate code-specific latent features conditioned on the textual description of each code. Detail of WGAN-GP is described in Appendix B. To condition on the code description, we use a label encoder function $C : \mathbb{L} \mapsto \mathbb{C}$ that maps the code description to a low-dimension vector $c$. We denote $c_l = C(l)$. The generator, $G : \mathbb{Z} \times \mathbb{C} \mapsto \mathbb{F}$, takes in a random Gaussian noise vector $z \in \mathbb{Z}$ and an encoding vector $c \in \mathbb{C}$ of a code description to generate a latent feature $\tilde{f}_l = G(z, c)$ for this code. The discriminator or critic, $D : \mathbb{F} \times \mathbb{C} \mapsto \mathbb{R}$, takes in a latent feature vector $f$ (either generated by WGAN-GP or extracted from real data examples) and the encoded label vector $c$ to produce a real-valued score $D(f, c)$ representing how realistic $f$ is. The WGAN-GP loss is:

$$\mathcal{L}_{\text{WGAN}} = \mathbb{E}_{(f,c) \sim P_\mathbb{S}^{f,c}}[D(f, c)] - \mathbb{E}_{(\tilde{f},c) \sim P_\mathbb{S}^{\tilde{f},c}}[D(\tilde{f}, c)] +$$
$$\lambda \cdot \mathbb{E}_{(\hat{f},c) \sim P_\mathbb{S}^{\hat{f},c}}[(||\nabla D(\hat{f}, c)||_2 - 1)^2] \tag{4}$$

where $(\cdot, c) \sim P_\mathbb{S}^{\cdot,c}$ is the joint distribution of latent features and encoded label vectors from the set of seen code labels $\mathbb{S}$, $\hat{f} = \alpha \cdot f + (1 - \alpha) \cdot \tilde{f}$ with $\alpha \sim \mathcal{U}(0, 1)$ and $\lambda$ is the gradient penalty coefficient. WGAN-GP can be learned by solving the minimax problem: $\min_G \max_D \mathcal{L}_{\text{WGAN}}$.

**Label encoder.** The function $C$ is an ICD-code encoder that maps a code description to an embedding vector. For a code $l$, we first use a LSTM (Hochreiter & Schmidhuber, 1997) to encode the sequence of $M$ words in the description into a sequence of hidden states $[e_1, e_2, \ldots, e_M]$. We then perform a dimension-wise max-pooling over the hidden state sequence to get a fixed-sized encoding vector $e_l$. Finally, we obtain the eventual embedding $c_l = e_l || g_l$ of code $l$ by concatenating $e_l$ with $g_l$ which is the embedding of $l$ produced by the graph encoding network. $c_l$ contains both the latent semantics of the description (in $e_l$) as well as the ICD hierarchy information (in $g_l$).

**Keywords reconstruction loss.** To ensure the generated feature vector $\tilde{f}_l$ captures the semantic meaning of code $l$, we encourage $\tilde{f}_l$ to be able to well reconstruct the keywords extracted from the clinical notes associated with code $l$.

For each input text $x$ labeled with code $l$, we extract the label-specific keyword set $K_l = \{w_1, w_2, \ldots, w_k\}$ as the set of most similar words in $x$ to $l$, where the similarity is measured by cosine similarity between word embedding in $x$ and label embedding $v_l$. Let $Q$ be a projection matrix, $\mathcal{K}$ be the set of all keywords from all inputs and $\pi(\cdot, \cdot)$ denote the cosine similarity function, the loss for reconstructing keywords given the generated feature is as following:

$$\mathcal{L}_{\text{KEY}} = -\log P(K_l | \tilde{f}_l) \approx - \sum_{w_k \in K_l} \pi(w_k, v_l) \cdot \log P(w_k | \tilde{f}_l)$$

$$= - \sum_{w_k \in K_l} \pi(w_k, v_l) \cdot \log \frac{\exp(w_k^\top \cdot Q\tilde{f}_l)}{\sum_{w \in \mathcal{K}} \exp(w^\top \cdot Q\tilde{f}_l)} \tag{5}$$

**Discriminating zero-shot codes using ICD hierarchy.** In the current WGAN-GP framework, the discriminator cannot be trained on zero-shot codes due to the lack of real positive features. In order to include zero-shot codes during training, we utilize the ICD hierarchy and use $f^{sib}$, the latent feature extracted from real data of the nearest sibling $l^{sib}$ of a zero-shot code $l$, for training the discriminator. The nearest sibling code is the closest code to $l$ that has the same immediate parent. This formulation would encourage the generated feature $\tilde{f}$ to be close to the real latent features of the siblings of $l$ and thus $\tilde{f}$ can better preserving the ICD hierarchy. More formally, let $c^{sib} = C(l^{sib})$, we propose the following modification to $\mathcal{L}_{\text{WGAN}}$ for training zero-shot codes:

$$\mathcal{L}_{\text{WGAN-Z}} = \mathbb{E}_{c \sim P_\mathbb{U}^c}[\pi(c, c^{sib}) \cdot D(f^{sib}, c)] - \mathbb{E}_{(\tilde{f}, c) \sim P_\mathbb{U}^{\tilde{f}, c}}[\pi(c, c^{sib}) \cdot D(\tilde{f}, c)] +$$

$$\lambda \cdot \mathbb{E}_{(\hat{f}, c) \sim P_\mathbb{U}^{\hat{f}, c}}[(||\nabla D(\hat{f}, c)||_2 - 1)^2] \tag{6}$$

where $c \sim P_\mathbb{U}^c$ is the distribution of encoded label vectors for the set of zero-shot codes $\mathbb{U}$ and $(\cdot, c) \sim P_\mathbb{U}^{\cdot, c}$ is defined similarly as in Equation 4. The loss term by the cosine similarity $\pi(c, c^{sib})$ to prevent generating exact nearest sibling feature for the zero-shot code $l$. After adding zero-shot codes to training, our full learning objective becomes:

$$\min_G \max_D \mathcal{L}_{\text{WGAN}} + \mathcal{L}_{\text{WGAN-Z}} + \beta \cdot \mathcal{L}_{\text{KEY}} \tag{7}$$

where $\beta$ is the balancing coefficient for keyword reconstruction loss.

**Fine-tuning on generated features.** After WGAN-GP is trained, we fine-tune the pretrained classifier $g_l$ from baseline model with generated features for a given zero-shot code $l$. We use the generator to synthesize a set of $\tilde{f}_l$ and label them with $y_l = 1$ and collect the set of $f_l$ from training data with $y_l = 0$ using baseline model as feature extractor. We finally fine-tune $g_l$ on this set of labeled feature vectors to get the final binary classifier for a given zero-shot code $l$.

## 4 EXPERIMENTS

### 4.1 SETUP

**Dataset description.** We use the publicly available medical dataset MIMIC-III (Johnson et al., 2016) for evaluation, which contains approximately 58,000 hospital admissions of 47,000 patients who stayed in the ICU of the Beth Israel Deaconess Medical Center between 2001 and 2012. Each admission record has a discharge summary that includes medical history, diagnosis outcomes, surgical procedures, discharge instructions, etc. Each admission record is assigned with a set of most

Table 1: Results on **seen codes** using ZAGRNN feature extractor described in Section 3.2

| Method | Micro | | | | Macro | | | |
|---|---|---|---|---|---|---|---|---|
| | Pre | Rec | F1 | AUC | Pre | Rec | F1 | AUC |
| ZAGRNN | 58.06 | 44.94 | 50.66 | 96.67 | 30.91 | 25.57 | 27.99 | 94.03 |
| ZAGRNN + $\mathcal{L}_{\text{LDAM}}$ | 56.06 | 47.14 | 51.22 | 96.70 | 31.72 | 28.06 | 29.78 | 94.08 |

Table 2: **Zero-shot** ICD coding results. Scores are averaged over 10 runs on different seeds.

| Method | Micro | | | | Macro | | | |
|---|---|---|---|---|---|---|---|---|
| | Pre | Rec | F1 | AUC | Pre | Rec | F1 | AUC |
| ZAGRNN | 0.00 | 0.00 | 0.00 | 89.05 | 0.00 | 0.00 | 0.00 | 90.89 |
| ZAGRNN + $\mathcal{L}_{\text{LDAM}}$ | 0.00 | 0.00 | 0.00 | 90.78 | 0.00 | 0.00 | 0.00 | 91.91 |
| ZAGRNN + Meta (Liu et al., 2019) | **46.70** | 0.89 | 1.74 | 90.08 | 3.88 | 0.95 | 1.52 | 91.88 |
| $\mathcal{L}_{\text{WGAN}}$ (Xian et al., 2018) | 23.92 | 17.63 | 20.30 | 91.94 | 17.30 | 17.38 | 17.34 | 92.26 |
| $\mathcal{L}_{\text{WGAN}} + \mathcal{L}_{\text{CLS}}$ (Xian et al., 2018) | 23.57 | 16.55 | 19.44 | 91.71 | **18.39** | 16.81 | 17.56 | 92.32 |
| $\mathcal{L}_{\text{WGAN}} + \mathcal{L}_{\text{CYC}}$ (Felix et al., 2018) | 23.97 | 17.93 | 20.51 | 91.88 | 17.86 | 17.83 | 17.84 | 92.27 |
| $\mathcal{L}_{\text{WGAN-z}} + \mathcal{L}_{\text{CLS}}$ | 22.49 | 17.40 | 19.62 | 91.80 | 16.56 | 17.26 | 16.90 | 92.16 |
| $\mathcal{L}_{\text{WGAN-z}} + \mathcal{L}_{\text{CYC}}$ | 21.44 | 17.24 | 19.11 | 91.90 | 16.05 | 17.06 | 16.54 | 92.25 |
| $\mathcal{L}_{\text{WGAN}} + \mathcal{L}_{\text{KEY}}$ (Ours) | 23.26 | 18.24 | 20.45 | 91.73 | 17.09 | 18.38 | 17.71 | 92.21 |
| $\mathcal{L}_{\text{WGAN-z}}$ (Ours) | 22.18 | 19.03 | 20.48 | 91.79 | 16.87 | 18.84 | 17.80 | 92.28 |
| $\mathcal{L}_{\text{WGAN-z}} + \mathcal{L}_{\text{KEY}}$ (Ours) | 22.54 | **19.51** | **20.91** | **92.18** | 17.70 | **19.15** | **18.39** | **92.34** |

relevant ICD-9 codes by medical coders. The dataset is preprocessed as in (Mullenbach et al., 2018). Our goal is to accurately predict the ICD codes given the discharge summary.

We split the dataset for training, validation, and testing by patient ID. In total we have 46,157 discharge summaries for training, 3,280 for validation and 3,285 for testing. There are 6916 unique ICD-9 diagnosis codes in MIMIC-III and 6090 of them exist in the training set. We use all the codes for training while using codes that have more than 5 data examples for evaluation. There are 96 out of 1,646 and 85 out of 1,630 unique codes are zero-shot codes in validation and test set, respectively.

**Baseline methods.** We compare our method with ZAGRNN modified from previous state of the art approaches on zero-shot ICD coding (Rios & Kavuluru, 2018) as described in Section 3.2, meta-embedding for long-tailed problem (Liu et al., 2019) and WGAN-GP with classification loss $\mathcal{L}_{\text{CLS}}$ (Xian et al., 2018) and with cycle-consistent loss $\mathcal{L}_{\text{CYC}}$ (Felix et al., 2018) that were applied to GZSL classification in computer vision domain. Detailed description and hyper-parameters of baseline methods are in Appendix C.

**Training details.** For WGAN-GP based methods, the real latent features are extracted from the final layer in the ZAGRNN model. Only features $f_l$ for which $y_l = 1$ are collected for training. We use a single-layer fully-connected network with hidden size 800 for both generator and discriminator. For the code-description encoder LSTM, we set the hidden size to 200. We train the discriminator 5 iterations per each generator training iteration. We optimize the WGAN-GP with ADAM (Kingma & Ba, 2015) with mini-batch size 128 and learning rate 0.0001. We train all variants of WGAN-GP for 60 epochs. We set the weight of $\mathcal{L}_{\text{CLS}}$ to 0.01 and $\mathcal{L}_{\text{CYC}}, \mathcal{L}_{\text{KEY}}$ to 0.1. For $\mathcal{L}_{\text{KEY}}$, we predict the top 30 most relevant keywords given the generated features.

After the generators are trained, we synthesize 256 features for each zero-shot code $l$ and fine-tune the classifier $g_l$ using ADAM and set the learning rate to 0.00001 and the batch size to 128. We fine-tune on all zero-shot codes and select the best performing model on validation set and evaluate the final result on the test set.

## 4.2 RESULTS

**Evaluation metrics.** We report both the micro and macro precision, recall, F1 and AUC scores on the zero-shot codes for all methods. Micro metrics aggregate the contributions of all codes to compute the average score while macro metrics compute the metric independently for each code and then take the average. All scores are averaged over 10 runs using different random seeds.

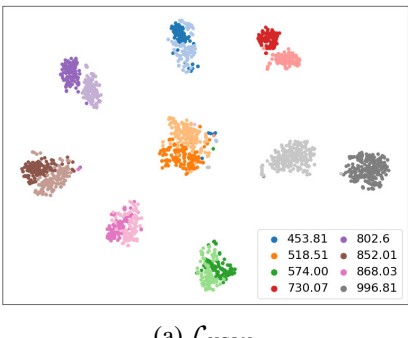 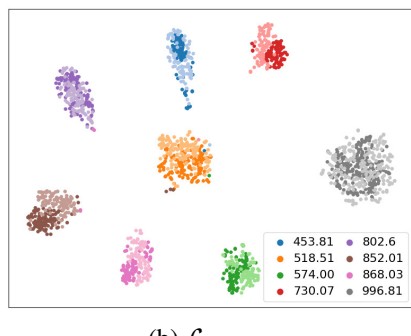

(a) $\mathcal{L}_{\text{WGAN}}$        (b) $\mathcal{L}_{\text{WGAN-Z}}$

Figure 3: T-SNE projection of generated features for zero-shot codes using (a) $\mathcal{L}_{\text{WGAN}}$ and (b) $\mathcal{L}_{\text{WGAN-Z}}$. Lighter color are projection of generated features and darker color are of real features from the nearest sibling codes. Features generated for zero-shot codes using $\mathcal{L}_{\text{WGAN-Z}}$ are closer to the real features from the nearest sibling codes. This shows that using $\mathcal{L}_{\text{WGAN-Z}}$ can generate features that better preserve the ICD hierarchy.

**Results on seen codes.** Table 1 shows the results of ZAGRNN models on all the seen codes. Note that fine-tuning the zero-shot code classifiers with meta-embedding or WGAN-GP will not affect the classification for seen codes since the code assignment classifiers are independently fine-tuned.

**Results on zero-shot codes.** Table 2 summarizes the results for zero-shot codes. For the baseline ZAGRNN and meta-embedding models, the AUC on zero-shot codes is much better than random guessing. $\mathcal{L}_{\text{LDAM}}$ improves the AUC scores and meta-embedding can achieve slighter better F1 scores. However, since these methods never train the binary classifiers for zero-shot codes on positive examples, both micro and macro recall and F1 scores are close to zero. In other words, these models almost never assign zero-shot codes at inference time. For WGAN-GP based methods, all the metrics improve from ZAGRNN and meta-embedding except for micro precision. This is due to the fact that the binary zero-shot classifiers are fine-tuned on positive generated features which drastically increases the chance of the models assigning zero-shot codes.

**Ablation studies on WGAN-GP methods.** We next examine the detailed performance of WGAN-GP methods using different losses as shown in Table 2. Adding $\mathcal{L}_{\text{CLS}}$ hurts the micro metrics, which might be counter-intuitive at first. However, since the $\mathcal{L}_{\text{CLS}}$ is computed based on the pretrained classifiers, which are not well-generalized on infrequent codes, adding the loss might provide bad gradient signal for the generator. Adding $\mathcal{L}_{\text{CYC}}, \mathcal{L}_{\text{KEY}}$ and $\mathcal{L}_{\text{WGAN-Z}}$ improves $\mathcal{L}_{\text{WGAN}}$ and achieves comparable performances in terms of both micro and macro metrics. At a closer look, $\mathcal{L}_{\text{WGAN-Z}}$ improves the recall most, which matches the intuition that learning with the sibling codes enables the model to generate more diverse latent features. The performance drops when combing $\mathcal{L}_{\text{WGAN-Z}}$ with $\mathcal{L}_{\text{CLS}}$ and $\mathcal{L}_{\text{CYC}}$. We suspect this might be due to a conflict of optimization that the generator tries to synthesize $\tilde{f}$ close to the sibling code $l^{sib}$ and simultaneously maps $\tilde{f}$ back to the exact code semantic space of $l$. Using $\mathcal{L}_{\text{KEY}}$ resolves the conflict as it reconstructs more generic semantics from the words instead of from the exact code descriptions. Our final model that uses the combination of $\mathcal{L}_{\text{WGAN-Z}}$ and $\mathcal{L}_{\text{KEY}}$ achieves the best performance on both micro and macro F1 and AUC score.

**T-SNE visualization of generated features.** We plot the T-SNE projection of the generated features for zero-shot codes using WGAN-GP with $\mathcal{L}_{\text{WGAN}}$ and $\mathcal{L}_{\text{WGAN-Z}}$ in Figure 3. Dots with lighter color represent the projections of generated features and those with darker color correspond to the real features from the nearest sibling codes. Features generated for zero-shot codes using $\mathcal{L}_{\text{WGAN-Z}}$ are closer to the real features from the nearest sibling codes. This shows that using $\mathcal{L}_{\text{WGAN-Z}}$ can generate features that better preserve the ICD hierarchy.

**Keywords reconstruction from generated features.** We next qualitatively evaluate the generated features by examining their reconstructed keywords. We first train a keyword predictor using $\mathcal{L}_{\text{KEY}}$ on the real latent features and their keywords extracted from training data. Then we feed the generated features from zero-shot codes into the keyword predictor to get the reconstructed keywords.

Table 3 shows some examples of the top predicted keywords for zero-shot codes. Even the keyword predictor is never trained on zero-shot code features, the generated features are able to find relevant words that are semantically close to the code descriptions. In addition, features generated with $\mathcal{L}_{\text{WGAN-Z}}$ can find more relevant keywords than $\mathcal{L}_{\text{WGAN}}$. For instance, for zero-shot code V10.62, the

Table 3: Keywords found by generated features using $\mathcal{L}_{\texttt{WGAN}}$ and $\mathcal{L}_{\texttt{WGAN-z}}$ for zero-shot ICD-9 codes. Bold words are the most related ones to the ICD-9 code description.

| Code | Description | Keywords from $\mathcal{L}_{\texttt{WGAN}}$ | Keywords from $\mathcal{L}_{\texttt{WGAN-z}}$ |
|---|---|---|---|
| V10.62 | Personal history of myeloid leukemia | AICD, inferoposterior, cardiogenic, **leukemia**, silent | **leukemia**, Zinc, **myelogenous**, **CML**, metastases |
| E860.3 | Accidental poisoning by isopropyl alcohol | apneic, pulses, choking, substance, fractures | **intoxicated**, **alcoholic**, AST, EEG, **alcoholism** |
| 956.3 | Injury to peroneal nerve | vault, **injury**, pedestrian, orthopedics, **TSICU** | **injuries**, **neurosurgery**, **injury**, **TSICU**, coma |
| 851.05 | Cortex contus-deep coma | **contusion**, **injury**, **trauma**, **neurosurgery**, **head** | **brain**, **head**, **contusion**, **neurosurgery**, **intracranial** |
| 772.2 | Subarachnoid hemorrhage of fetus or newborn | **subarachnoid**, **SAH**, neurosurgical, screening | **subarachnoid**,**hemorrhages**, **SAH**, **newborn**, **pregnancy** |

Table 4: **Few-shot** ICD coding results. Scores are averaged over 10 runs on different seeds.

| | Micro | | | | Macro | | | |
|---|---|---|---|---|---|---|---|---|
| Method | Pre | Rec | F1 | AUC | Pre | Rec | F1 | AUC |
| ZAGRNN | **64.00** | 1.27 | 2.48 | 92.11 | 4.15 | 1.23 | 1.90 | 90.99 |
| ZAGRNN + $\mathcal{L}_{\texttt{LDAM}}$ | 60.53 | 1.82 | 3.53 | 92.10 | 6.29 | 1.80 | 2.80 | 90.74 |
| ZAGRNN + Meta (Liu et al., 2019) | 48.88 | 6.75 | 11.84 | 92.15 | 16.65 | 6.77 | 9.62 | 90.92 |
| $\mathcal{L}_{\texttt{WGAN}}$ (Xian et al., 2018) | 29.18 | 18.14 | 22.37 | 92.59 | 20.76 | 18.09 | 19.33 | 90.99 |
| $\mathcal{L}_{\texttt{WGAN}}$ + $\mathcal{L}_{\texttt{CLS}}$ (Xian et al., 2018) | 29.18 | 17.67 | 22.01 | 92.54 | 19.88 | 17.62 | 18.68 | 91.01 |
| $\mathcal{L}_{\texttt{WGAN}}$ + $\mathcal{L}_{\texttt{CYC}}$ (Felix et al., 2018) | 28.82 | 18.43 | 22.48 | 92.57 | 20.39 | 18.28 | 19.28 | 90.96 |
| $\mathcal{L}_{\texttt{WGAN-z}}$ + $\mathcal{L}_{\texttt{CLS}}$ | 27.97 | 17.70 | 21.68 | 92.58 | 20.18 | 17.59 | 18.80 | 91.01 |
| $\mathcal{L}_{\texttt{WGAN-z}}$ + $\mathcal{L}_{\texttt{CYC}}$ | 28.40 | 18.24 | 22.22 | 92.61 | 20.82 | 18.17 | 19.40 | 90.99 |
| $\mathcal{L}_{\texttt{WGAN}}$ + $\mathcal{L}_{\texttt{KEY}}$ (Ours) | 28.97 | 18.31 | 22.44 | 92.62 | 20.92 | 18.24 | 19.49 | **91.05** |
| $\mathcal{L}_{\texttt{WGAN-z}}$ (Ours) | 27.66 | 18.81 | 22.39 | 92.56 | 20.45 | 18.81 | 19.59 | 90.97 |
| $\mathcal{L}_{\texttt{WGAN-z}}$ + $\mathcal{L}_{\texttt{KEY}}$ (Ours) | 27.95 | **18.96** | **22.60** | **92.63** | **21.55** | **18.92** | **20.15** | 91.00 |

top predicted keywords from $\mathcal{L}_{\texttt{WGAN-z}}$ include *leukemia, myelogenous, CML (Chronic myelogenous leukemia)* which are all related to myeloid leukemia, a type of cancer of the blood and bone marrow.

**Results on few-shot codes.** As we have seen promising results on zero-shot codes, we also evaluate our feature generation framework on few-shot ICD codes, where the number of training data for such codes are less than or equal to 5. We apply the exact same setup as zero-shot codes for synthesizing features and fine-tuning classifiers for few-shot codes. There are 220 and 223 unique few-shot codes in validation and test set, respectively.

Table 4 summarizes the results. The performance of ZAGRNN models on few-shot codes is slightly better than zero-shot codes yet the recall are still very low. Meta-embedding can boosts the recall and F1 scores from baseline models. WGAN-GP methods can further boosts the performance on recall, F1 and AUC scores and the performance using different combination of losses generally follows the pattern in zero-shot code results. In particular, our final model trained with $\mathcal{L}_{\texttt{WGAN-z}}$ and $\mathcal{L}_{\texttt{KEY}}$ can perform slightly better than other WGAN-GP models in terms of F1 and AUC scores.

## 5 CONCLUSION

We introduced the first feature generation framework, AGMC-HTS, for generalized zero-shot multi-label classification in clinical text domain. We incorporated the ICD tree hierarchy to design GAN models that significantly improved zero-shot ICD coding without compromising the performance on seen ICD codes. We also qualitatively demonstrated that the generated features using our framework can preserve the class semantics as well as the ICD hierarchy compared to existing feature generation methods. In addition to zero-shot codes, we showed that our method can improve the performance on few-shot codes with limited amount of labeled data.

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

## A    APPENDIX: GATED RECURRENT UNITS

Below is the detailed construction of `GRUCell` in equation 1 from Section 3.2:

$$z_l^t = \sigma(W_z \cdot h_l^t + U_z \cdot g_l^{t-1} + b_z)$$
$$r_l^t = \sigma(W_r \cdot h_l^t + U_r \cdot g_l^{t-1} + b_r)$$
$$g_l^t = (1 - z_l^t) \odot g_l^{t-1} + z_l^t \odot \tanh(W_h \cdot h_l^t + U_h \cdot (r_l^t \odot g_l^{t-1}) + b_h)$$

where $\odot$ is the dimension-wise multiplication.

## B    APPENDIX: GENERATIVE ADVERSARIAL NETWORKS

GANs (Goodfellow et al., 2014) have been extensively studied for generate highly plausible data. The idea of GAN is to train a generator and a discriminator through a minimax game. The generator takes in a random noise and generate fake data to fool the discriminator while the discriminator tries to distinguish between generated data and real data. The training procedure of GANs can be unstable, thus Arjovsky et al. (2017) proposes Wasserstein-GAN (WGAN) to counter the instability problem by optimizing the Wasserstein distance instead of the original Jenson-Shannon divergence. Gulrajani et al. (2017) further improves WGAN by using gradient instead of weight clipping for the required 1-Lipschitz constraint in WGAN discriminator.

## C    APPENDIX: MORE TRAINING DETAILS

**ICD-9 code information.** We extract the ninth version of the ICD code descriptions and hierarchy from the CDC website[1]. In addition to the official description, we extend the descriptions with medical knowledge, including synonyms and clinical information, crawled from online resources[2].

**ZAGRNN.** For the ZAGRNN model, we use 100 convolution filters with a filter size of 5. We use 200 dimensional word vectors pretrained on PubMed corpus[3] (Zhang et al., 2019b). We use dropout on the word embedding layer with rate 0.5. We use the ADAM (Kingma & Ba, 2015) for optimization with a minibatch size of 8 and a learning rate of 0.001. The final feature size and GRNN hidden layer size are both set to 400. We train the ZAGRNN model for 40 epochs.

**Meta-embedding.** Liu et al. (2019) proposed meta-embedding for solving large long-tail problem by transferring knowledge from head classes to tail classes. The method naturally fits ICD coding due to the long-tailed code distribution. To apply meta-embedding in ICD coding, we first construct a set of centroids $M$ as the mean of $f_l$ for each code $l$ from the training data. Let $\odot$ denote dimension-wise multiplication, then the meta-embedding for $f$ is calculated as:

$$f^{meta} = f + e(f) \odot (o(f)^\top \cdot M) \tag{8}$$

where $o(f)$ is the attention scores for selecting centroids $M$ and $e(f)$ is a dimension-wise coefficient for selecting the attended features. Both $o$ and $r$ are parameterized as neural networks and are learned during fine-tuning. The final classification is performed by $\hat{y}_l = \sigma(g_l^\top \cdot f_l^{meta})$.

For meta-embedding, we fine-tune the neural network modules $e$ and $o$ using ADAM and set learning rate to 0.0001 and batch size to 32.

**WGAN-GP with classification loss.** Xian et al. (2018) proposed to add a cross-entropy loss during training WGAN-GP to generate features being correctly classified as conditioned labels. In ICD coding, this loss translates to enforcing $\tilde{f}$ being classified as positive for code $l$:

$$\mathcal{L}_{\text{CLS}} = -\log P(y_l = 1|\tilde{f}) = -\log \sigma(g_l^\top \cdot \tilde{f}_l) \tag{9}$$

**WGAN-GP with cycle consistency loss.** Similar to adding $\mathcal{L}_{\text{CLS}}$ to prevent the generated features being random, Felix et al. (2018) proposed to add a loss that constrains the synthetic representations to generate back their original semantic features. Let $R : \mathbb{F} \mapsto \mathbb{C}$ be a linear regression estimate the label embedding $c_l$ from the generated feature $\tilde{f}_l$, the cycle consistency loss is defined as:

$$\mathcal{L}_{\text{CYC}} = ||c_l - R(\tilde{f}_l)||_2^2 \tag{10}$$

---

[1] https://www.cdc.gov/nchs/icd/icd9cm.htm
[2] http://www.icd9data.com/
[3] https://github.com/ncbi-nlp/BioWordVec

## D    APPENDIX: PRECISION-RECALL CURVES

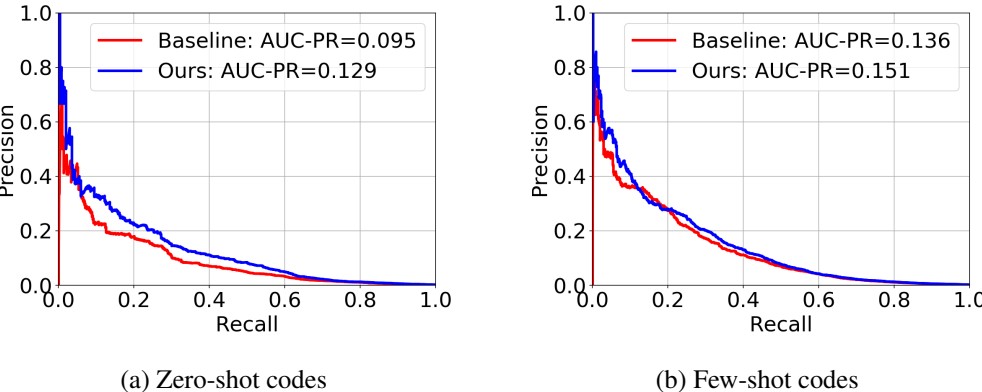

(a) Zero-shot codes                    (b) Few-shot codes

Figure 4: Micro averaged precision-recall curves comparisons on zero-shot codes (a) and few-shot codes (b). Red line is the baseline feature extractor described in Section 3.2 and blue line is from our final model. The legend denotes the correspond area under the curve (AUC) value.

We plot the precision-recall curve with micro averaged scores for previous state of the art ICD coding model and ours on few-shot codes and zero-shot codes. As demonstrated in Figure 4, our model can achieve better precision and recall trade-off and also higher area under the curve scores for both zero-shot codes and few-shot codes.

## E    COMPARISON OF FEATURE EXTRATORS ON SEEN ICD CODES

Table 5: Results on all the **seen codes** using different feature extractors described in Section 3.2

| Method | Micro | | | | Macro | | | |
|---|---|---|---|---|---|---|---|---|
| | Pre | Rec | F1 | AUC | Pre | Rec | F1 | AUC |
| ZAGCNN (Rios & Kavuluru, 2018) | **58.29** | 44.64 | 50.56 | 96.59 | 30.00 | 24.65 | 27.06 | 94.00 |
| ZAGRNN (ours) | 58.06 | 44.94 | 50.66 | 96.67 | 30.91 | 25.57 | 27.99 | 94.03 |
| ZAGRNN + $\mathcal{L}_{\text{LDAM}}$ (ours) | 56.06 | **47.14** | **51.22** | **96.70** | **31.72** | **28.06** | **29.78** | **94.08** |

We compare the performance of our modified ZAGRNN with the original ZAGCNN proposed by Rios & Kavuluru (2018) on seen ICD codes in Table 5. Discussion of the modification is detailed in Section 3.2. With ZAGRNN, almost all metrics slightly increased from ZAGCNN except for micro precision. With $\mathcal{L}_{\text{LDAM}}$ loss, our final feature extractor can improve more significantly from ZAGCNN especially for macro metrics and achieve better precision recall trade-off. These modifications are not enough to get reasonable performance zero-shot codes as shown in Table 2, mainly due to the lack of positive example for zero-shot codes during training.

