# OpenReview forum: "Generalized Zero-shot ICD Coding"
_ICLR.cc/2020/Conference — Reject_

### Official Review · AnonReviewer1 · 2019-10-23
**Official Blind Review #1**

**Rating:** 3

**Review:**

The paper deals with the problem of text classification when the number of class is large (17000) and most of the classes do not have examples in the training set. This problem is known as Zero-shot learning. The paper proposes to use adversarial methods and the hierarchical organisation of the classes to improve current models.

The paper lacks a clear description of the complete system : Figure 1 seems to be the one but there is no mention of the classification part. The description of each bloc is clear enough independently  but many question remains on the global picture.

The author should for example emphasis the fact that in their problemn the classes come with a short text  description. This is somehow unususal for a text classification problem where usually the classes are not defined by a description. In the figure, the difference of the processing of the text from the clinical document and from the class description is not clear.

3.1 is related to feature extraction for the clinical document, but it is also said that this processing is also apply to class description.

3.2 label encoder : if the notations were the same as on figure A, it would help understanding.

Results : the claim "our methods improve the F1 score from nearly 0 to 20.91% for the zero-shot codes" is not true : state-of-the-art models such as Xian2018 and Felix2018 are already over 20% F1.

When looking at Figure 3 and the analysis, WGAN-Z seem to provide better representation that WGAN with it has almost no impact on the classification results (F1 20.48 versus F1 20.30)

In conclusion, it seems that the proposed methods make the model more complex without bringing a significant improvements.

**Experience Assessment:**

I have read many papers in this area.

**Review Assessment: Checking Correctness Of Derivations And Theory:**

I assessed the sensibility of the derivations and theory.

**Review Assessment: Checking Correctness Of Experiments:**

I assessed the sensibility of the experiments.

**Review Assessment: Thoroughness In Paper Reading:**

I read the paper at least twice and used my best judgement in assessing the paper.

---

> ### Author Response · Authors · 2019-11-15
> **Response to Reviewer #1**
>
> We thank Reviewer 1 for the insightful comments and would like to address the specific questions below.
>
> Comment #1. Description of the complete system:
> Thanks. We have added more detailed description of the complete system in the revised version. Please kindly refer to the second paragraph of Section 1 and Section 3.1 for a description of our complete system and global picture, including classification part. Detailed discussion on classification was included in the “Multi-label classification” paragraph of Section 3.2 and “Fine-tuning on generated features” paragraph of Section 3.3.
>
> Comment #2. Emphasis that the classes come with a short text description:
> Thanks. We have added the emphasis of this fact in both Section 1 and Section 3.1 in the revision.
>
> Comment #3. “3.1 is related to feature extraction... is also apply to class description.”
> As explained in “Label-wise feature extraction” paragraph of Section 3.2 in the revised version (3.1 in the previous version), we process the label description in the feature extractor so as to extract label-wise features for each document. For an input document, a total of |L| feature vectors are extracted and each feature vector contains the relevant information to each label description in the set of labels L.
>
> Comment #4. “3.2 label encoder : if the notations... help understanding.”
> Thanks. We have updated figure 1 with the Label encoder. Other notations in the figure are consistent with the text.
>
> Comment #5. “Results : the claim ‘our methods improve the F1 score from nearly 0 to 20.91% for the zero-shot codes’ is not true”
> We would like to point out that Xian2018 and Felix2018 are published works on image classification in the computer vision research community, not on ICD coding. The only existing baseline on the generalized zero-shot ICD coding is Rios and Kavuluru’s work and their F1 score is nearly zero. We are the first to propose the adversarial generative model with hierarchical label structure for the GZSL text classification problem and achieved ~21% F1.
>
> Even compared with Xian2018 and Felix2018, our final model has achieved significant improvement. We have performed student-t test on micro F1 scores. The p-values are 0.00018 and 0.014 comparing to Xian2018 and Felix2018, indicating the improvement between our final model and prior works are statistically significant.
>
> Comment #6. “When looking at Figure 3 and the analysis... impact on the classification results (F1 20.48 versus F1 20.30)”
> We conducted student-t test on micro F1 scores from WGAN-Z and WGAN and the p-value is 0.01 which demonstrates that the improvement of WGAN-Z is statistically significant.
> The goal of Figure 3 is to show that features generated for zero-shot codes using WGAN-Z are closer to the real features from the nearest sibling codes, which indicates that using WGAN-Z can generate features that better preserve the ICD hierarchy.
>
> Comment #7. “It seems that the proposed methods make the model more complex without bringing a significant improvements”.
> As mentioned above, via student-t test, we verified that the new modules proposed in our model yield statistically significant improvement over the baselines adapted from Xian2018 and Felix2018. Xian2018 and Felix2018 focused on image classification, instead of ICD coding in multi-label text classification, therefore they did not address several key challenges in zero-shot ICD coding: 1) The class label distribution is extremely long-tailed and there are many codes without labeled data. 2) The label space has complex hierarchical tree structure. 3) The input clinical documents are very long and noisy and the key information corresponding to each assigned code is quite sparse.
>
> To solve these challenges, it is necessary to propose new ideas. Specifically, we proposed a novel conditional generative model with hierarchical label structure, which has three novel modules to address the three challenges.  1) our framework adversarially generates features for zero-shot codes and finetune the classifiers with the synthetic features, 2) we exploit the hierarchical tree of the ICD codes by encouraging the features of zero-shot codes to be similar to their siblings, 3) we reconstruct the code-relevant keywords in the input documents from the synthetic features to guarantee the semantic consistency.
>
> More importantly, via significance tests, each of these proposed modules has demonstrated to be effective in improving ICD-coding performance. These modules together frame our novel approach, which, to the best of our knowledge, is the first adversarial generative model with hierarchical label structure for generalized zero-shot ICD coding problem. Besides, compared with the only existing baseline on the generalized zero-shot ICD coding (Rios and Kavuluru’s work), we improve the F1 score from nearly zero to ~21%.

---

### Official Review · AnonReviewer2 · 2019-10-24
**Official Blind Review #2571**

**Rating:** 6

**Review:**

This paper addresses the problem of zero-shot prediction for ICD codes for medical notes. The main idea of the paper is to use GANs to generate latent features for the text conditioned on the ICD codes and taking into account the tree-like structure of the ICD codes themselves. The classification of ICD codes is trained on the GAN-extracted features.

The input clinical document is represented as a concatenation of its word embeddings. The representation of an ICD code is realized by averaging the embeddings of the words in its corresponding description. The feature extraction model takes the word embeddings from the input text and passes them through a CNN; in addition, attention is used for the encodings of all the ICD codes.

For further encoding the ICD codes, the tree-like organization of the codes is exploited by using GRNNs. The generated features together with the graph-encodings of the ICD codes are used to classify the clinical text.

Some suggestions for improvement:

* It was not clear the relationship with previous work by Rios and Kavuluru. I had to refer to the paper to be able to appreciate the differences. The feature extraction network seems to be similar (Rios & Kavuluru used GCNNs instead of GRNNs to encode the ICD codes). The GAN-based feature generation seems to be new in this paper.

* I did not understand the comment regarding the independent training of the zero-shot cases vs the rest.

* Siblings ICD codes mean codes that share an immediate parent?

* Table 1. seems to cite the related work results, but it doesn't seem to include the results of the proposed method.

* In Table 2, for the cases where precision and recall are 0, are the AUC numbers correct? Table 2 also shows the results for the ablation studies that are discussed later in the section. It took me a while to understand the difference between Table 2 and Table 4.

* For Table 4, you could add the explanation on what you considered few-shot cases for those results.

I think the paper could improve the explanation of the components of the model and the presentation of the evaluation results.

I think the idea of using GANs to generate features that can help with the classification + taking advantage of the relationship/structure among the ICD codes are both interesting ideas.

**Experience Assessment:**

I have published one or two papers in this area.

**Review Assessment: Checking Correctness Of Derivations And Theory:**

I assessed the sensibility of the derivations and theory.

**Review Assessment: Checking Correctness Of Experiments:**

I carefully checked the experiments.

**Review Assessment: Thoroughness In Paper Reading:**

I read the paper thoroughly.

---

> ### Author Response · Authors · 2019-11-15
> **Response to Reviewer #2**
>
> We thank Reviewer 2 for the constructive suggestions and sincerely appreciate your positive feedback on the novelty and effectiveness of the proposed methods.  We would like to address the specific comments below.
>
> Comment #1. Relationship with previous work by Rios and Kavuluru:
> Thanks for your detailed comment and we made it clearer about the relationship with the previous work by Rios and Kavuluru in the revised version (uploaded in OpenReview). Please kindly refer to the discussion of the limitations of Rios and Kavuluru’s work in Section 2, and the discussion of the difference between ours and Rios and Kavuluru’s work in the first paragraph of Section 3.2. We only use the modified Rios and Kavuluru’s model to serve as the feature extractor and pretrained classifier for our proposed approach. As acknowledged by Reviewer 2, the modifications include: 1) replace the graph convolutional networks with graph recurrent neural network; 2) adopting the label-distribution-aware margin loss for training. The performance gain from our modification is added to Table 5 in Appendix E.
>
> Besides the above modification, we would like to highlight other differences of our framework to Rios and Kavuluru’s: 1) our framework adversarially generates features for zero-shot codes to finetune the classifiers. 2) We exploit the tree structure of ICD codes by encouraging the synthetic features of zero-shot codes to be similar to their siblings 3), we reconstruct the code-relevant keywords in the input documents from the synthetic features to guarantee the semantic consistency.  To conclude, we are the first to propose the adversarial generative model with hierarchical label structure for the generalized zero-shot text classification problem and the whole framework is brand new compared with Rios and Kavuluru’s work.
>
> Comment #2. Clarify the comment regarding the independent training of the zero-shot cases vs the rest:
> By independent training we mean that we only fine-tune the binary classifiers for zero-shot codes with the synthetic features while keeping the pretrained classifier for the rest of the codes fixed. Please kindly refer to the explanation in the last paragraph of Section 3.1 and Section 3.3, and “results on seen codes” paragraph of Section 4.2.
>
> Comment #3. Definition of siblings ICD codes:
> Sibling codes are codes that share an immediate parent. Since each code can have more than one sibling, we select the nearest sibling code measured by the cosine similarity in the encoded label space. Please refer to “Discriminating zero-shot codes using ICD hierarchy” paragraph in Section 3.3 for the detailed explanation.
>
> Comment #4. Results in Table 1:
> Thanks for your detailed comments and we are sorry for any confusion caused by the citation. Table 1 shows the performance of our modified feature extractor on the SEEN codes and these performance will remain the same after fine-tuning on zero-shot codes as we do not finetune on SEEN codes (explained in the last paragraph of Section 3.1 and Section 3.3, and “results on seen codes” paragraph of Section 4.2). As explained for Comment #1, we modified Rios and Kavuluru’s work to serve as the feature extractor and pretrained classifier for our generative framework. The first row in Table 1 shows the results for modified ZAGRNN, and the second row shows the results for modified ZAGRNN trained with label-distribution-aware margin loss. We will clarify this in the final version.
>
> Comment #5. In Table 2, for the cases where precision and recall are 0, are the AUC numbers correct?
> Precision and recall are 0 because the baseline model assign a very low probability to zero-shot codes and thus never predicts these codes on any input text given the default threshold of 0.5. AUC can be larger than the random guessing baseline of 0.5 as AUC is measured with all possible thresholds.
>
> Comment #6. Explanation on what you considered few-shot cases for those results.
> Thanks for your suggestion. Please find our detailed explanation about few-shot cases in the last paragraph “Results on few-shot codes” of Section 4.2.
>
> Comment #7. I think the paper could improve the explanation of the components of the model and the presentation of the evaluation results.
> Thanks for your suggestion. We will carefully improve the explanation of the model components and the presentation of the evaluation results in the final version.

---

### Official Review · AnonReviewer3 · 2019-11-04
**Official Blind Review #3**

**Rating:** 6

**Review:**

The paper proposes a method to do zero-shot ICD coding, which comes down to determining which elements from a set of natural language labels (ICD codes) apply to a given text (diagnostic summary). The main problem is that many codes have zero or very few examples. The proposed solution for this problem is to learn a feature-space generator for examples which can be conditioned on a code. This generator is trained using a GAN, which moves the few-/zero-shot problem to training the discriminator. Here, the tree structure of the ICD codes is used, by using examples with sibling labels as approximate examples for codes with few or on labelled data points. Additionally, a keyword reconstruction loss is used, based on the idea that the keywords of the corresponding ICD code can be reconstructed from a good feature vector.

The paper is written and executed well; the setup, which involves a fair number of components, is described and motivated clearly. The experiments include comparisons to state of the art results, finding that applying the GAN-based technique they choose (introduced in Xian et al., 2018 for computer vision problems) produces significant gains in recall of low-data classes (few or zero examples). The precision decreases, probably due to a shift from false negatives to false positives, while the AUC shows modest gains.

The main potential issue with the paper is the degree of (effective) novelty. The bulk of the gain relative to the SotA seems to be achievable by applying the GAN-based method, which is described in the Xian et al. (2018) reference, or by applying the label-distribution-aware margin, which are known techniques even if they have not been applied to this particular dataset yet. The additional elements introduced by the authors - the use of sibling codes and the keyword reconstruction loss - are good ideas, and it is worthwhile having a documented test of the benefit they provide, but they don’t seem to have a major influence on the quality of the model.

All in all, I would argue for the paper to be accepted. The work done here is valuable to have on record, and the presentation and execution are well done.

Two questions for the authors:

1. Could the model benefit from having a self-attention model, like a transformer? This applies mostly to the diagnostic text encoder, as interpreting the ICD code descriptions seems not to depend strongly on structure or context. From the text of the paper it appears that a 1D convolution was used to process the diagnostic texts, but I would expect that longer-distance links between words can be quite relevant there.

2. Could the precision-recall curves be added to the supplementary material?

**Experience Assessment:**

I do not know much about this area.

**Review Assessment: Checking Correctness Of Derivations And Theory:**

I assessed the sensibility of the derivations and theory.

**Review Assessment: Checking Correctness Of Experiments:**

I assessed the sensibility of the experiments.

**Review Assessment: Thoroughness In Paper Reading:**

I read the paper at least twice and used my best judgement in assessing the paper.

---

> ### Author Response · Authors · 2019-11-15
> **Response to Reviewer #3**
>
> We thank Reviewer 3 for the insightful comments. We appreciate your positive feedback and address the specific questions below.
>
> Comment #1 & #2. Degree of (effective) novelty:
> We would like to emphasize that prior works on generalized zero-shot learning (GZSL) mainly focused on visual tasks, the study of GZSL for multi-label text classification is largely under-explored.  Generalized zero-shot ICD coding is a much harder task and we cannot simply apply GAN-based method as in vision problem (e.g., Xian et al. 2018) or label-distribution-aware margin loss to solve it, due to the following aspects: 1) The label distribution is extremely long-tailed and it is challenging to perform multi-label classification on both seen codes and zero-shot codes. 2) The label space has complex hierarchical tree structure. 3) The input clinical documents are very long and noisy and the key information corresponding to each assigned code is quite sparse.
>
> To solve the above challenges, we proposed a novel conditional generative model with hierarchical label structure. Our model has three new components to address the aforementioned three challenges. For the first challenge, our framework adversarially generates features for zero-shot codes to finetune the classifiers. For the second challenge, we exploit the tree structure of ICD codes by encouraging the features of zero-shot codes to be similar to their siblings. For the third challenge, we reconstruct the code-relevant keywords in the input documents from the synthetic features to guarantee the semantic consistency. These three model components were not contained in the model of Xian et al. (2018).
>
> Each module contributed to the improved performance, and all of them together frame our novel approach, which, to the best of our knowledge, is the first adversarial generative model with hierarchical label structure for generalized zero-shot ICD coding problem.
>
> Regarding the effectiveness, we have performed student-t test comparing our final model against previous GAN based approach on micro F1 scores. The p-values are 0.00018 and 0.014 comparing to Xian2018 and Felix2018, indicating the differences between our results and prior works are statistically significant.
>
> Question #1. Could the model benefit from having a self-attention model, like a transformer?
> The clinical documents are very long and each document can have thousands of tokens.  We use 1D convolution for feature extraction mainly because it is most efficient to train for long text. Current state of the art transformer models (BERT, GPT-2) are all trained on sentence level inputs and are extremely memory consuming for longer sequences[1], mainly due to the n x n self-attention matrix in each layer where n is the number of tokens. It is an interesting future direction to explore whether we can perform efficient training and feature extraction with more advanced encoder models like transformers on long clinical texts.
>
> Question #2. Could the precision-recall curves be added to the supplementary material?
> Thanks for your constructive suggestion. We added the Precision-recall curves to Appendix D of the revised version (uploaded in OpenReview).
>
> Reference
> [1] https://github.com/google-research/bert/blob/master/README.md#out-of-memory-issues

---

### Author Response · Authors · 2019-11-15
**List of changes in the revised version**

Dear reviewers,
We uploaded a revised version of our paper and below is the list of changes based on the suggestions from the reviews.

Reviewer #1, #2, #3
We revised the introduction to emphasize the background of generalized zero-shot ICD coding problem and the novelty of our framework.

Reviewer #1
We modified Figure 1 and added Label Encoder to keep it consistent with the text.
We revised and added a subsection 3.1 as an overview and global picture of our framework.

Reviewer #2
We clarified definition of nearest sibling in Section 3.3.

We slightly revised the experimental section to make the explanation of each experiment more organized. We removed the misleading citation and caption in Table 1, which shows the results of our modified feature extractor on seen ICD codes. We added the comparison of our modified feature extractor with the original work [1] in Table 5 of Appendix E. We emphasized the captions in Table 2 and 4 so that the difference is clearer.

Reviewer #3
We added the precision-recall curves in Appendix D.

Reference:
[1] Rios and Kavuluru, Few-Shot and Zero-Shot Multi-Label Learning for Structured Label Spaces, EMNLP 2018

---

### Decision · Program_Chairs · 2019-12-19

**Decision:**

Reject

**Comment:**

This paper proposes a method to do zero-shot ICD coding, which involves assigning natural language labels (ICD codes) to input text. This is an important practical problem in healthcare, and it is not straightforward to solve, because many ICD codes have none or very few training examples due to the long distribution tail. The authors adapt a GAN-based technique previously used in vision to solve this problem. All of the reviewers agree that the paper is well written and well executed, and that the results are good. However, the reviewers have expressed concerns about the novelty of the GAN adaptation step, and left this paper very much borderline based on the scores it received. Due to the capacity restrictions I therefore have to recommend rejection, however I hope that the  authors resubmit elsewhere.